# Can Asset-Based Community Development with Children and Youth Enhance the Level of Participation in Health Promotion Projects? A Qualitative Meta-Synthesis

**DOI:** 10.3390/ijerph16193778

**Published:** 2019-10-08

**Authors:** Rita Agdal, Inger Helen Midtgård, Vigdis Meidell

**Affiliations:** Western Norway University of Applied Sciences, 7030, 5020 Bergen, Norway; inger.helen.midtgard@hvl.no (I.H.M.); vigdis.meidell@hvl.no (V.M.)

**Keywords:** community-based participatory research, health promotion/methods, child advocacy, child, adolescent, public health

## Abstract

The asset-based community development (ABCD) approach have been widely used to map local assets and to ensure participation of local communities in public health promotion strategies. Participatory practices, such as ABCD, have been applied to shift public health strategies towards addressing health inequities. In this meta-synthesis, we ask if, and how, ABCD enhance the level of participation for children, youth and schools. Three thousand eight hundred eight titles and abstracts were identified in ten databases and transferred to the online program Rayyan. Through a blinded process we excluded texts that did not meet the inclusion criteria. The twelve included texts on ABCD for children, youth and schools are of varying quality. The research on ABCD for children, youth and schools have not been cumulative. Nevertheless, the texts show that ABCD provides strategies that enhance the participation of children, youth, and schools, in health promotion projects. The projects were categorized according to Robert Hart’s classical participation ladder, and we found that the projects with the highest level of adherence to ABCD principles also had the highest level of participation. The projects with high levels of participation were supported by adult facilitators that created learning environments where children and youth developed their participatory skills.

## 1. Introduction

Asset-based community development (ABCD) is one of several approaches employed in health promotion strategies to engage communities, focusing on establishing networks and collective action [1,2]. As a move to shift public health practices towards addressing health inequalities and inequities, community involvement and participatory practices have become more central to health promotion strategies the last decades [3,4]. Some see this as a re-orientation of public health strategies, while others see it as a return to “the old public health” [5,6]. To involve and strengthen local communities and to ensure community action for health is a stated aim in the Ottawa charter [7], which was a milestone for health promotion. The importance of developing healthy and sustainable communities to reduce health inequalities was highlighted in the Marmot review [8]. The review concluded that a broader focus is needed in health promotion projects, and suggested that the communities, community groups and the third sector should be engaged in health promotion to identify challenges and solutions. Community engagement may involve both direct and indirect pathways to reduce health inequalities. Such approaches have been found to be effective in reducing inequalities in health, and a review suggests that public health initiatives should incorporate community engagement approaches into intervention design [9].

There has been a growing interest in ABCD the recent years from central health authorities that aim to develop health promotion strategies that involve the communities, for example, in Norway [1,4,10,11], United Kingdom and in Denmark [12]. It has been seen as a way to address health inequality by involving communities, and as a move towards a more positive asset-based focus for public health strategies targeting health and well-being, away from the disease prevention model and the focus on mortality, morbidity and disability [1]. ABCD has been seen as an approach to reduce the impact of the social determinants of health and reduce health inequality and inequity, through community engagement. Although several approaches to involve communities, ABCD is the only strategy that suggests a focus on local assets and strengths only [1]. It is one of the strength-based approaches that have been employed to involve children, youth and the educational sector in health promotion strategies [13,14].

### ABCD Strategies

The first ABCD handbook “Building communities from the inside out. A path towards finding and mobilizing community assets” [15] resulted from research by John Kretzmann and John McKnight. They visited communities and observed that some of them had mobilized their own assets, despite challenges and low incomes, whereas, other communities believed that improvements only could come with the help of outside experts. After this observations, Kretzmann and McKnight suggested that local development would be more likely to succeed if it is strength-based and focused on local capacities, rather than to be deficit-driven and focused on needs ([15], p. 13). Kretzmann and McKnight warned against the experts’ tendencies to be deficit-focused, rather than working with local people to mobilize their assets and define their own needs and visions ([15], p. 1–5). They suggest that an ABCD process can be defined by three characteristics ([15], p. 9):

Citizen led. Local citizens map their resources and needs and lead the collaboration with outside partners.

Relationship oriented. There is a focus on building social networks.Asset-based. The process focusses on strengths, resources, and assets.

The handbook, which the ABCD Institute refers to the book as “the basic manual”, is a practical guide with suggestions on how to mobilize citizens and communities, to map assets and to promote partnerships to create the community the inhabitants envision. In two dedicated sections, the book describes how children, youth and schools can be included as agents in community renewal processes. They describe how to release individual capacities with youth ([15], pp. 29–46), and how to involve schools ([15], pp. 206–224) and community colleges ([15], pp. 227–240) as local institutions for community building.

The handbook describes how ABCD could be used as a strategy, and later others have added their interpretations of ABCD. Through a realist synthesis, Blickem et al. [16] set out to map how ABCD was understood and practiced. They identified variations of the three basic characteristics, and the use of appreciative inquiry. Appreciative inquiry is also used in other settings, and has inspired some of the methods used for asset mapping, in line with the focus on strengths and empowerment [12,16].

The ABCD handbook inspired worldwide activity, but has been criticized along several lines, both regarding lack of clarification and for neglecting issues related to power. It has been pointed out that the ABCD approach relies on concepts from economic language, like “asset” and “deficit” and “development”, which is not clearly defined [6]. One could also argue that the “asset” concept needs to be flexible, not to limit how local citizens may identify their own “personal assets” and “communal assets” as they do during the ABCD process. The open exploration of local assets, by the community, frequently leads to the “Aha!” moments that are characteristic of ABCD, where people discover themselves as capable of creating changes [17]. The openness that creates enthusiasm locally may, however, challenge external agents with more limited agendas and those responsible for evaluating outcomes. Evaluation can become a difficult task when initiators refrain from clearly defining outcomes from the outset [18]. Others have argued that the evaluation methodology needs to be adapted to the complexity of the projects [9,16]. Some authors have been concerned that policy makers who aim to reduce public budgets [19,20] may employ ABCD strategically to pave the way towards reducing budgets. It should not be ignored that ABCD is used within the political context of austerity, but how it affects policies remains to be studied, and whether policies would change without the presence of ABCD is a different question. Harrison et al. [21] point out that ABCD may also enables the optimal use of resources that are still available, by involving citizens and their local knowledge [21]. There are few long term studies of ABCD [16], and the influence of ABCD on policies, and what ABCD contributes to producing under different circumstances has not been thoroughly explored, as far as we know. Another line of critique against ABCD has been that it may reinforce power relations, locally and between community groups and external agents [19]. We do not reject the critique, but note that so far this claim is based on theoretical works and not on empirical observations. On the contrary, the main body of ABCD research points to processes of engagement and empowerment, where local groups gain more influence over their living conditions, as their assets become visible and a resource for the community and social networks strengthened. This is also reported in projects involving children and youth [14]. We can sum up that ABCD has gained momentum over the years, and that various researchers have pointed to limitations and conditions that may limit the success of ABCD approaches.

Harrison et al. [21] point out that there is limited evidence regarding what population the approach is suitable for. We have not identified reviews on ABCD used by or with children and youth. One article states that it is a review of the internatiaonal literature on school health outcomes using the framework of ABCD [22]. However, this review includes studies of a public health intervention in schools in general, and merely employs ABCD as a lens to shed light on alternative ways to approach public health work. It concludes that an ABCD approach may help national governments develop resources for education and health, and that schools can be used as assets ([22], p. 13). Another text is published as a review article on ABCD and youth mental health [23], but is not an actual review of the literature. It discusses the importance of social capital to promote mental health in general and refers to the use of ABCD.

Twenty-six years after the handbook was published, we seek to sum up what can be learned from the practice and research on ABCD for children and youth, and how that can be used to increase their participation in health promotion projects. As ABCD in many cases engages local communities, some projects involve youth and children without being specific. At the outset, we anticipated that we would find studies of the participation of young people and children in ABCD processes that took place in their communities and where they were target groups. We expected that the basic characteristics of ABCD would be present, as found by Blickem et al. in their general realist review [16]. We found the ABCD approach interesting as it could be an approach to increase the participation of children and youth in health promotion projects both in a long and short term. One of the short term wins when engaging participation of children and youth in health promotion projects is to improve the adaptation of health promotion measures to their needs by their active participation, i.e., Reference [24]. In addition to providing ways to increase participation, and mapping of what they see as their personal resources and what they see as communal resources, we expected the approach to increase integration and social networks and in addition contribute to various kinds of social and participatory competence for the future. Such results would in itself be a realization of some of the aims stated in the Ottawa charter of health promotion [7]. To investigate whether the studies of ABCD with children and youth would be practiced in line with the basic characteristic and produce such positive results was for us an open-ended question.

We will in this article review literature on ABCD employed in projects with or by children and youth, and ask if, and how, ABCD enhance the level of participation. As the participation of children and youth have been a challenge [2,25] in health promotion projects, we intend to explore whether ABCD may be a relevant approach. Before presenting how strategies of ABCD are employed in projects with or by children and youth, we present the methods that were applied in the review and meta-analysis. After a brief presentation of the included case descriptions, we discuss how the projects described in the included texts adhere to three basic characteristics of the ABCD described in the original handbook [15,26]. Secondly, we discuss the forms of participation described in the texts on ABCD projects, and thereafter we move on to discuss how adherence to the ABCD principles may be related to the level of participation. Finally, we return to our primary research question; Can ABCD enhance the participation of children and youth in health promotion projects?

## 2. Classification of Participation

The perception of children and youth and their capability to participate varies across contexts and cultures. Still, children and youth are frequently perceived as citizens in the making, rather than being entitled to active citizenship [27]. Thus, to engage children and youth in health promotion, and in any type of project, involve particular conditions related to both perceptions of their roles and their participatory skills. It has been pointed out that the status of childhood is only there for a short time for each individual, so that the possibility for continuous building of collective organizations or competence is limited. That it is rare to find grass-root movements consisting of children also means that youth and children representatives may have fragile legitimacy. It also implies that the representatives who do not have an organisation or movement that support them may be more susceptible to replicate the power structures and to accept agendas set by others [27]. The participation of children and youth has been seen as critical in public health interventions, for various reasons, as, for example, to increase the effectiveness of disease prevention or lifestyle changes [19,24], but also to establish accountability, ownership, and empowerment [2], as mediators to reach goals related to inequality and related to the community based goals of the Ottawa charter [7].

There are many ways to increase the participation of youth and children in health promotion projects, for example, based on; peer-teaching, behavior planning frameworks, diffusion of innovation, self-determination theory, positive youth and life development frameworks, the four R’s model, YPAR, photo voice and asset mapping. Some methods are mainly used to strengthen interventions aimed at individual prevention of disease, whereas, other methods aim to involve children and youth in defining both the problem and possible solutions. Scholars have presented typologies of children and youth participation, also for health promotion [2]. As ABCD is gaining popularity in general and has been recommended as a strategy to engage children and youth [15,17,22,23], it is timely to examine how ABCD can improve participation beneficial to health promotion projects. However, an understanding of the level of the participation of children and youth, and how this comes about, should be based on an analysis of the literature on ABCD projects. To analyze the available studies with regard to participation, we need to identify specific forms of participation. We will, thus, relate the included texts to Robert Hart’s classical ladder of children’s participation [28] as this is well known and has more classes than most others, including that of Wong et al. [2].

We focus on categorizing the participation as it is described in the articles on ABCD with children, youth and schools, related to Hart’s ladder. It must be noted that Hart and others have warned against using the ladder as an evaluation tool, as not all projects aim at a high level of participation. Still, the ladder may serve as a metaphor and provide the specificity needed to categorize different levels of participation and for a critical discussion of children’s participation [25]. The three lowest rungs show apparent involvement, described as manipulation, decoration and tokenism. Real participation starts at rung four, where children are thoroughly informed about the project and who controls it. At this level, the children participate voluntarily and are given meaningful, functional roles, such as guides at events. On the highest levels, rung seven and eight, adults do not participate. On rung eight, children initiate the projects to work on issues that they themselves have identified, and share decisions with adults ([28], pp. 11–14). According to Hart ([25], pp. 21–22), the ladder was intended as a tool to help bring a critical perspective to a subject that lacked one, and to evoke (self-) consciousness and dialogue about child participation. We will use the ladder as a starting point for categorization and discussion of the texts on ABCD projects which involves children, youth or schools.

It has been argued that Harts typology ignores that the highest levels of participation are not always feasible, and that non-hierarchical typologies should be used, i.e., Reference [29]. Hart [25] has, however, clarified that his typology is a tool to describe forms of participation and stimulate dialogue, and not a prescription. Recent attempts to develop more nuanced and flexible frameworks, still refer to the basic typology from Hart. In one of the latest attempts to re-conceptualize youth participation it is stated that “there is a need for framework which focuses on the fluid nature of participation, with its ongoing responses to context, circumstances and to the shifts in relational power that can involve, ebb and flow within a given venture ([30], p. 247)”. We agree that terminology development is valuable, but we find that the original typology introduced by Hart [28] suffices for our review purpose. Thus, we will classify the participation of children and youth in the ABCD project according to Hart’s categories.

## 3. Methods and Materials

Through an initial scoping exercise to prepare the systematic search, conducted by the first and second authors IHM and RA, we found that the research on ABCD mostly has been qualitative. We prepared to include both quantitative and qualitative studies, and to synthesize the qualitative research findings pertaining to ABCD and participation for children and youth. However, we did not identify any quantitative studies on ABCD. We have used the PRISMA checklist as far as possible, although not all items on the checklist are relevant for qualitative studies. We developed a search string, as can be seen in Table 1, and adapted it to the following databases for the period 1.1.1998 to 25.9.2018: Cinahl, Scopus, Book citations Scopus, Book citations WoS, Web of Science, Embase, Eric, Medline, PsycInfo, SocIndex.

We included terms for “community” as this is a central term in ABCD approaches, although the “community” in the literature sometimes is a group that is not localized the same place, in combination with terms related to the approach as “asset” or “strength” based. We included the term “appreciative inquiry”, as this has been perceived as a fundament for the asset mapping process in ABCD [12,31]. The search string did not limit the search to youth and children, as we wanted to go through the full texts about ABCD projects to see if children or youth were included in more of the texts on ABCD.

We included all texts published in English, German, Danish, Swedish and Norwegian, and sorted the 3808 titles using the online program Rayyan. The first two authors removed duplicates and read titles and abstracts to exclude articles from the total search that did not meet the inclusion criteria. This part of the process was blinded, as the two first authors, read through the full search result independently, using Rayaan, to decide if the texts focused on ABCD projects. Articles were first screened in the blinded process to decide if they met the first criteria: They should either refer to ABCD as the practical approach, or refer to the original handbook of ABCD as the fundament for the approach.

We then used the “un-blind” function in Rayyan to compare the decisions suggested by the first and the second author and found conflicting suggestions for four papers. This was resolved by a discussion between the two first authors, whereupon we reached an agreement. Secondly, we read the 119 papers on ABCD to identify texts that included children, schools or youth, no older than twenty-five years. The limit of twenty-five was set to potentially include studies with mixed age groups in schools. We did, however, not identify studies with this wide age range. The oldest youth in the studies were upper secondary school pupils. A search in the reference lists of the included articles did not provide additional articles that fit the inclusion criteria. We did not do a full citation review analysis.

We systematically identified how the projects adhered to the three principles for ABCD and the level of participation as described in the ladder of children’s participation. This was done by the two first authors who systematically read the texts related to the three basic characteristics of ABCD, first individually, and then discussed and compared their judgments. The two authors agreed, but found that the information in three of the texts were scarce. The third author, VM, took part in the discussion of the levels of participation. The two first authors (R.A. and I.H.M.) evaluated the 12 included papers according to the EPICURE elements [32].

### Some Methodological Challenges in the Included Texts

Although ABCD is a widespread approach, we only identified 12 studies focusing on school, children and youth, including one unpublished report. R.A. and I.H.M. evaluated the included papers according to the EPICURE elements suggested by Stige et al. [32]. This is an approach to the evaluation of heterogenous, rich and substantive accounts [32]. EPICURE is an acronym proposing attention to the reflexive dialogue, rather than a checklist-based judgement. Following EPICURE we evaluated the following elements for each of the included text: Engagement of the researcher and the relationship to the situation studied; Processing of the empirical material; Interpretation of the material; Critique of the merits and limits of research; Usefulness and value for practice; Relevance for the discipline(s), and Ethical considerations. We found that the majority of the texts have methodological flaws related to the EPICURE elements: Most authors of these studies conceal their roles in the projects and do not provide precise descriptions of facilitation, inclusion criteria and processes. This information is relevant to the evaluation of the quality of the studies, and also for the discussion of whether the projects were in line with the basic principles of ABCD. Stige et al. [32] emphasize the importance of reflexivity regarding engagement as a quality criterion in qualitative research, and this is missing in many of the included studies [33,34,35,36]. For example, Gulley [35] does not describe her own role in the project, but refers to “Ms G” as the central initiator in charge of the project, without revealing any more information about her. She does not show how (if) the participants had any influence on the project. The aim and processes leading to the choice of focus and of participants are also lacking in some studies [35,36,37,38]. Only four of the articles [14,33,34,39] were explicit about how specific schools and youth were recruited. The eABCD article by Shah [36] is of an overall high standard, according to the EPICURE values [32], but it is not clear as to whether the students and the children voluntarily participated in the project or if it was mandatory. Only 5 of the articles [14,34,36,40,41] report on how data was processed or analyzed, or on alternative interpretations. Despite the flaws in quality, we find the studies relevant and useful.

## 4. Results

We included 12 texts that describe ABCD projects with youth, children and schools, from a total of 119 texts on ABCD, as illustrated in the PRISMA Flow Diagram in Figure 1. 11 text are peer-reviewed publications where, as mentioned, the quality of the descriptions varies, and one is an unpublished report. The report was included because it is substantial and sheds light on our research question, and because the total number of texts is relatively low. One text focus on pre-school children. Four studies focus on elementary school children. Four texts report on high school youth. Five texts describe “service-learning projects”, where the primary aim was for students to practice how to utilize ABCD targeting children and youth. Four of the latter had clearly defined health promotion goals, and one project was mainly conducted online, as “e-ABCD”. The included texts can be seen in Table 2.

### 4.1. Pre-School Children

One project targeted obesity amongst pre-school children [38]. Families, communities, local care systems, schools and other public and private interests initiated activities for children and families preceding, during and following two designated TV turn off weeks” ([38], p. 437). The authors conclude that ABCD is an effective way to mobilize communities in public initiatives ([38], p. 437), but the role of the children is not described.

### 4.2. Elementary School Children

Six studies report on projects with elementary school children [14,33,34,35,37,42]. The first, describes that children aged four to seventeen participated in the painting of a mural together with adults, and that the process produced social capital [35]. The second describes a substance abuse prevention project, where school children from a public housing project, made videos [33]. Together with the children, the facilitators invited participation by local associations, community leaders, parents and locals who were in recovery processes, partly to connect the children with positive role models. They developed knowledge about substance abuse and prevention based on their own interviews with locals, and they attained video-producing skills, self-esteem and leadership skills. The community engagement made it possible to adapt the drug education content to the particular community ([33], p. 315). The facilitators sought to enhance community ownership of the program, empower the community, use community capacities and to establish it as an ongoing program ([33], p. 315). In the third project, 56 children established their own project after participating in an ABCD project with adults [14]. A skilled facilitator participated. After three years, the group numbered 100 members. They developed skills in communication, performing arts, community service, literacy, negotiation, leadership, self-governing at a community level, empathizing and doing something for others. They also gathered knowledge about the local history and culture, and learned about ways to advance the quality of life in their communities and to address gaps in community assets for children ([14], p. 206). In the fourth study, ABCD contributed to the collaboration between adult guardians, villages chiefs and the MCM staff to strengthen caretakers, whereas, children were an indirect target group in this initial phase ([34], p. 15).

Two texts report on the program “Communities and Physicians Together” (CPT) [37,42] where communities and a pediatrics residency program collaborate has collaborated since 1999 on health promotion for children by the use of ABCD. The program includes grass-root leaders, NGOs, health centers and schools. In one case [37], a student noticed that children in one of the communities were frequently bitten by dogs. By talking to the children, she found that they had been unintentionally provoking the dogs due to a lack of knowledge of canine behavior. The student and the children planned and implemented a dog-safety fair with the community ([37], p. 1187). Another student noted that many children had to provide snacks and dinner for themselves and that they chose unhealthy food [42]. She engaged the children in an after-school program in collaboration with a local grocer, and together they developed a cookbook with healthy recipes. Pan et al. report that ABCD has increased social capital for children and families, and mobilized communities to improve social environments for children, thereby leading to improved health ([37], p. 1187).

### 4.3. High School Youth

Five texts report on three different projects with high school youth [36,39,40,41,43]. The pilot project reported by Roberts et al. [39] was initiated by a junior high school and aimed to discover, connect and mobilize the assets of students, and to connect them with assets in the local community. Throughout one school term, 15 grade 9 students met weekly for one-half day, in addition to afternoons, some full-day workshops and special events and projects. The initial project leaders and facilitators were from the Community Development Office. The youth chose projects to work on with community members, based on their own interests and assets ([39], p. 5). In addition to meeting the objectives, the project contributed to increased self-esteem, leadership capabilities, sense of responsibility to the local community, critical thinking skills, including identifying challenges associated with community asset mapping, and the importance of teamwork ([39], p. 56). The participants became more connected to their communities and took on active roles. The students were intrigued by the economic diversity and the resources they found in the communities, and expressed that they would use asset mapping again. Two articles by Andresen [40,41] report from a project where university students involved youth and children in asset mapping. Three communities had asked for assistance to facilitate dialogues to promote revitalization and development ([43], pp. 285–286). Youth in the project answered surveys in the form of post cards, and youth made a “participatory video” where they interviewed peers about their community. Later the youth set guidelines for child and adult participation in planning. Ten high school students participated in photography classes and prepared nature photography display boards ([41], p. 59). At the end of the ABCD project, 78% participants of all ages reported improved perception of the community. In the third project, students set out to improve digital learning for local youth by employing “eABCD” [36]. Dialog on a digital platform that was transformed into a partnership between the students and local youth [36]. The students demonstrated strategies that can be used to enact the themes of ABCD digitally in online service-learning and to counter approaches that are based on identifying deficits rather than strengths.

### 4.4. Degrees of Adherence to ABCD Principles and Levels of Participation

The adherence to the three basic ABCD principles varies, and so do the participation of children and youth. The participation varies in ways that can be categorized from the lowest to the highest rungs of the participation ladder. This is illustrated in Figure 2, and we describe the differences in the next paragraphs.

### 4.5. Non-Participation and Low Degrees of Adherence to ABCD Principles

The projects with the lowest degrees of participation also had the lowest adherence to the ABCD principles. The participation level in three projects corresponds to what Hart described as non-participation [28], as they only included children as target groups [34,38], and not as participants. One of the projects was, however, in an early phase ([34], p. 15) and seemed to aim for increased participation in the next phase. Another health promotion project emphasized the involvement of Reference [38] families, communities, local care system, schools and other public and private interest groups, but do not seem to have consulted the children ([38], p. 437) who later became service users. The text [38] nevertheless refer to ABCD as a useful strategy for collaboration between local groups from the community and professionals for health promotion. In the third of these projects, the children were given little freedom, but could choose which part of a mural to paint ([35], p. 24). This can be described as tokenism, referring to the third-lowest rung.

### 4.6. Informed Participation and Partial Adherence to ABCD Principles

Five of the projects, described in six texts, correspond to the fifth rung on the participation ladder; they are adult initiated, consulted and informed [37,39,40,41,42,43]. The youth and children who participated in these projects volunteered to do so and were free to express themselves within the pre-decided frames. Children and youth in these projects gradually became more engaged and developed ownership.

The service-learning projects do not clearly adhere to the first principle of ABCD, as local citizens did not initiate them. The local citizens did, however, recognize the challenges that external experts suggested that they work on, and took an active part in creating solutions. The children and youth in the service-learning programs [34,36,37,42,43], involved other community members in finding solutions. Service-learning projects serve to educate future ABCD facilitators and to strengthen partnerships between educational institutions and communities. The texts on the service-learning projects do not describe how facilitation of participation for children and youth were followed up. These projects lead to positive changes, and ABCD added to the learning process of students, while connecting educational institutions with local communities, children and youth. These projects may have brought out even more of the potential of ABCD if a facilitator had involved the children from the start, enhanced the focus on local assets from the start, and let them define the focus for local development and which challenges to focus on. That outsiders define the aims for projects will, however, be an unyielding premise for many ABCD -inspired health promotion projects, as external initiators and funding often arrive because external experts have defined local challenges. The problem of external experts defining local challenges is not a new discussion in health promotion (i.e., Reference [44]).

Initially, the project described by Andresen [40,41] was not designed to be relationship-building or internally driven, but became so when local inhabitants took ownership over the mapping in the survey, in line with the first and second ABCD principle. Nevertheless, only some children understood the purpose of the project, many children did not engage more than they were told to in the project. In adherence to the second principle, the included service-learning projects were relationship-building as they, to some extent, involved collaboration between local partners [34,37,40,41,42,43]. According to Shah [36], the e-project connected local youth and college students and showed that it is possible to build relationships by using eABCD, although it is not clear if they lasted after the service-learning project is not clear ([36], p. 208).

### 4.7. High Level of Participation and Full Adherence to the ABCD Principles

The three projects with the highest level of children’s participation [14,39,45] also adhere to the three basic principles of ABCD. The participation corresponds to rung six to eight. In two of the projects, the children were inspired by the adults’ ABCD project and initiated their own, as described by. The children first learned about the ABCD approach from other projects where multiple generations contributed. This knowledge was part of the fundament that enabled them to lead their own ABCD process, assisted by skilled facilitators. These projects raised awareness about community assets, as well as their personal assets, and they became agents of change for their communities [14,39] Johnsen Butterfield et al. [14] point out, that ABCD was used to address the needs of all children, including those who had learning or mental health challenges, or disabilities, which could disrupt classroom learning [14].

In the third text that describes a project with a high level of participation [33], adults initiated an approach to prevent drug use, but the level of the children’s participation gradually increased. The children became engaged because challenges related to substance abuse already were familiar to them. They gradually took on more leading roles as their self-esteem and competence rose, and they reached decisions regarding the content of the drug prevention videos. A team of residents, including youth, was trained to run the program to make sure that it could be continued without external experts ([33], p. 305). The involvement of community leaders and family members played an important part in mobilizing the youth and supporting the continuity of their engagement ([33], p. 314).

All three projects involved experienced adult facilitators who provided guidance, and to some extent, supportive parents and community leaders. Facilitators cautiously supported the participants with a focus on developing their participatory skills and on raising awareness of their own skills and gifts. Whereas, ABCD has been criticized for a general lack of focus on power relations [46], these three projects involved awareness of friction and inequality. The awareness of possible conflicts and inequalities seem to have strengthened both the individual and collective competence and agency, as when, for instance, the children chose to reach out and include other groups in actions of solidarity. ABCD projects also have been criticized for aligning with neoliberal values, where citizens are forced to take on tasks that used to be solved by the state [47]. However, in these three projects, we note that children and youth developed more awareness of the needs of others [14,33,39]. They brought up issues such as gender discrimination and poverty, aimed to improve the conditions for the whole community [14], and became more “responsive to responsibility” [39].

## 5. Discussion

### May the ABCD Approach Enhance Participation in Health Promotion Initiatives?

When discussing if ABCD can enhance participation in health promotion projects for children and youth, we must take the quality of the ABCD projects into consideration. The projects with the highest degree of participation adhere to the ABCD principles and employ skilled facilitators. Most of the fifteen texts describe projects that did not fully adhere to the principles, and might potentially have increased the level of participation by utilizing them. In a recent text, one of the ABCD founders emphasize that the three characteristics should be present when the process is described as ABCD ([26], p. 2). The handbook may, however, have been ambiguous about the ABCD approach, as it emphasizes flexibility and states that ABCD is “not (a) cookie cutter set of solutions”. This implies that the projects which do not adhere to the principles should be referred to as “inspired by” ABCD, rather than ABCD-based [26]. The major shortcoming, according to the ABCD principles, of the majority of projects, is that they are not citizen-initiated or citizen-led. Our findings suggest that when the goals are set by experts, the asset mapping activities and mobilization activities tend to be limited to a previously specified goal, hampering the potential to include local knowledge. The importance of including local knowledge as a resource in health promotion projects have been emphasized, both to ensure relevant projects, that they are efficient, and to foster empowerment and resilience [44,48]. In the reviewed texts on children and youth and schools, we find that those that start with an open-ended mapping of assets and then set the goals based on these assets leads to a more strength-based process where relationships, as well as competence, develop locally. In a realist synthesis on the application of ABCD, the authors find that the three characteristics are presents in the projects, but do not consider their order [16]. The original handbook states that “the local people must take the first step” ([15], p. 376) and suggests that community building must start with mapping community assets ([15], p. 345).

The projects with a high level of participation had skilled facilitators who based their work on the ABCD principles. The facilitator role is not thoroughly described, but when external experts initiate projects, Kretzmann and McKnight ([15], p. 351) suggest that they start by asking the community to define their own goals and assets. When the projects do not adhere to this, they are not firmly rooted in the local communities, and thus, do not utilize the potential of local knowledge. This makes it more challenging to adhere to the next two principles. In a recent paper one of the founders John McKnight, and Cormac Russel, emphasize that the order is critical. Assets should firstly be mapped by citizens, who in turn, decide on what action to take ([26], p. 6). The citizens’ roles are likely to be reduced if the mapping is done after deciding on the aim. When the process starts with asset mapping, including individual and collective assets, these assets can be mobilized and linked to build the community and the goals they set. In the projects with a high level of participation, the children and youth mapped both personal and communal resources early in the process. The mapping of resources seems to have an impact on their engagement and their sense of empowerment. Particularly the mapping of their own resources may have provided empowerment, as they discover their “power to”, whereas, the young normally have relatively little “power over”. Wong et al. discuss that this “power to” in many cases creates illusions, as youth may have little or no influence over-development ([2], p. 107). However, when the development is based on the assets of those who participate, it creates a possible fundament for their continued influence.

Some researchers’ have been concerned that ABCD may promote neoliberal values [47], or values of individualization and privatization ([20], p. 430). On the contrary, the participation promoted through ABCD has also been discussed as a possibility to support active citizen roles in a collectivist tradition, including for children and youth, as a contrast to consumer roles associated with neoliberal values [47]. This may be a simplistic distinction between types of citizen roles, but the children and youth developed values through these projects that challenge notions of individualization and privatization (i.e., References [14,33,39]). The included text does, however, point to the development of empathy with others, awareness of poverty, inequity and discrimination, critical awareness, solidarity, and responsiveness [14,33,39]. This empowerment is linked to both individual and collectivistic processes, where active citizenship can be developed alongside participatory skills. Children in high-income countries tend to spend more time in age-segregated groups in activities structured by adults, during the school day, as well as in after-school programs and other spare-time activities. Arenas for free play and self-organized activities are also arenas where children and youth practice their participation skills across age groups.

The ABCD approach proposes strategies to strengthen the balance of partnerships between professionals and citizens, and between adults, youth and children. The dependence on external facilitators will, however, make the projects vulnerable, as other projects, although the bottom-up approach is a fundament for the facilitator to lead by stepping back. This approach implies a different role for external experts, as facilitators and “connectors”, who engage with children and youth. Most of the included texts portray the projects as harmonious, with a few exceptions. Potential challenges related to power relations, co-production, and inequality are hardly present in the included texts or the handbook [15]. Although the role of the facilitator was poorly described in the handbook, it has later been described in relation to some of the successful projects (i.e., Reference [14], pp. 205–206). To develop awareness around the facilitator role seems crucial, as the deliberate attempt to lead by stepping back is contrary to what most professionals have learned. Funding is often tied to agendas, which challenges the open process whereby groups set their own agenda. Some of the health promotion projects that applied ABCD seem to utilize the engagement and adaption that is possible by the approach (i.e., References [37,38,42]). Although the level of participation was medium, the projects engaged and built relations both locally and between organizations, that could be a base for increased resilience and community agency. By facilitation, professionals may enable communities to form collectives for actions that have additional long-term value, compared to “solving problems”. As Forrester et al. [49] point out, a balance for government is required between setting strategic direction and facilitating the space for local self-determination. When children and youth in communities map their assets and decide on the aims, they establish a fundament for further community-driven development that builds participatory competence, relationships, and the communities themselves.

## 6. Conclusions

Research on ABCD for children, youth, and schools, is scarce, and of varying quality, therefore no final conclusions can be drawn. Nevertheless, the texts included in this review show that ABCD provides useful strategies to enhance the participation of children and youth in health promotion projects. The projects with the highest levels of adherence to ABCD principles had the highest level of participation, as categorized by means of the participation ladder [28]. The projects with high levels of participation were supported by adult facilitators who created good learning environments where children and youth could develop participatory skills. Several studies conclude that ABCD led to increased participation by children and youth in social networks in their local communities, active participation in local development work, and increased social capital. The children and youth developed an awareness of local assets, both personal and collective, awareness of different groups and interests in the community, increased sense of connectedness and a positive identity related to the community. The children and youth who initiated their own projects did so after they participated in adult-led ABCD projects, where they learned about the approach and developed participatory skills. Children and youth learned about various kinds of participation, experienced empowerment, increased self-esteem, developed skills related to leadership, communication, negotiation, critical thinking, mobilization and network-building skills. In addition, some projects led to increased literacy, and the participants practiced skills like interviewing, how to make videos, photography, to make exhibitions, and other practical skills.

The ABCD approach provided ways to adapt public health promotion projects to fit community characteristics and laid the groundwork for further community development, with or without external support. We found that partnerships were formed on an individual and collective level, and between organizations. Educational organizations were in some projects driving partners and provide resources, both when using ABCD as a basis for service-learning, including use of online “eABCD”, by providing places to meet, skilled staff who could facilitate processes with children and youth, and by strengthening partnerships between local and external partners.

Clarifications and more precise descriptions of the role of facilitators and external experts would be useful for further development of the ABCD approaches. Although ABCD is used worldwide, there are few studies with scientific rigor and precise descriptions of the processes and outcomes. Caan [22] has suggested that more quantitative studies on ABCD are required, and we would add that more qualitative studies are needed to describe processes and results. Additional studies based on triangulation of qualitative and quantitative methods are also needed to understand and describe processes and to relate them to outcomes. As with other complex interventions, it is a challenge to measure the effects of ABCD, which are often long term and affected by other internal and external influences on community development. As health authorities and policy makers consider comprehensive use of ABCD, studies on the relation between ABCD practices and policies would be useful.

## Figures and Tables

**Figure 1 ijerph-16-03778-f001:**
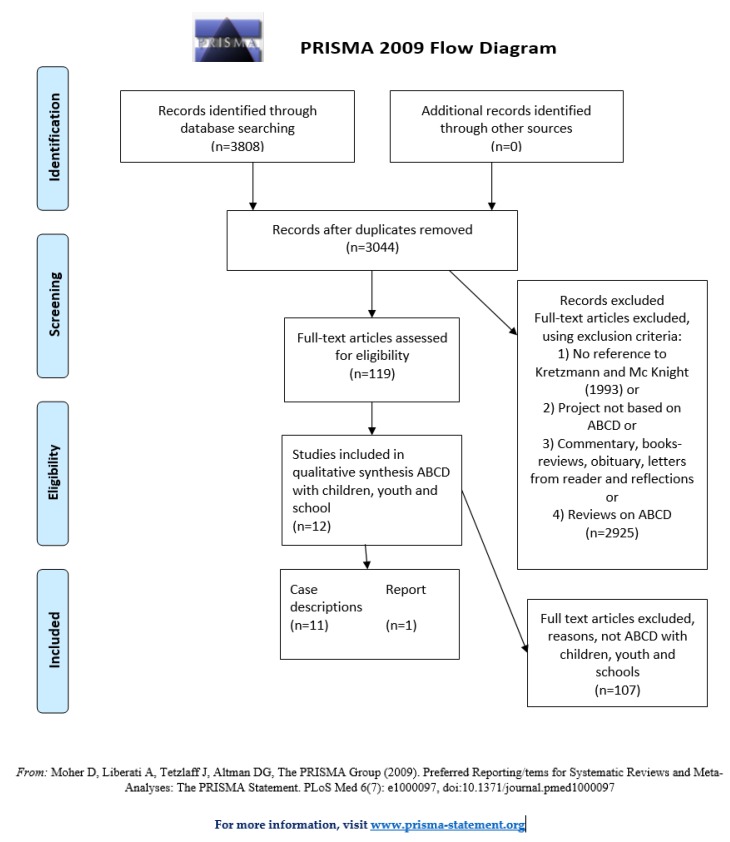
PRISMA flow diagram. ABCD, asset-based community development.

**Figure 2 ijerph-16-03778-f002:**
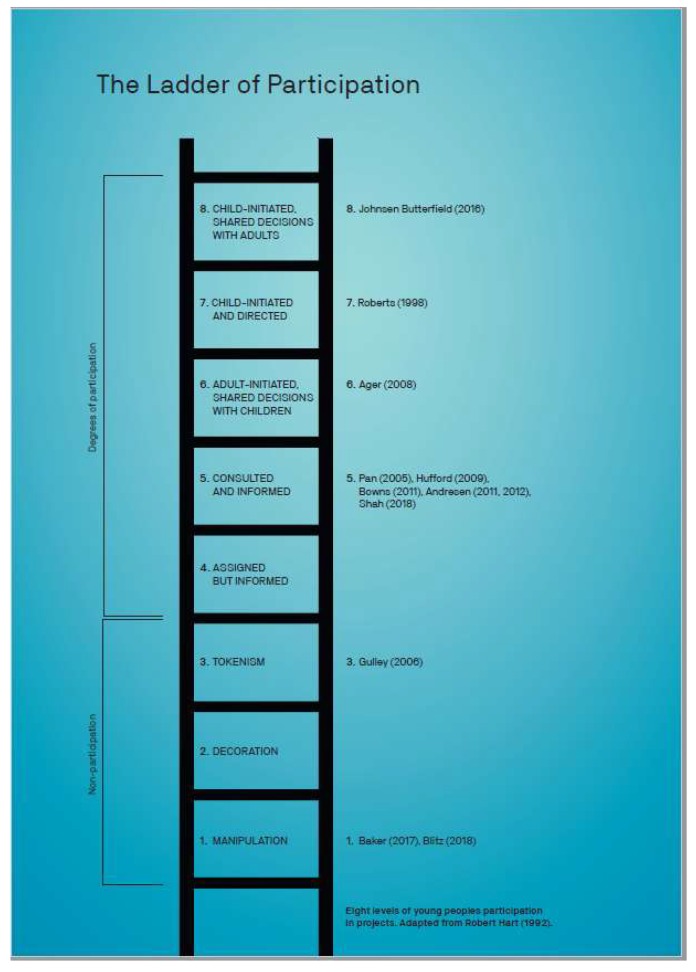
Adherence to ABCD principles and level of participation.

**Table 1 ijerph-16-03778-t001:** The search string.

#	Query
S3	S1 AND S2
S2	(community or communities or city or cities or town* or village* or neighborhood* or neighborhood* or urban or rural)
S1	(“asset*-based” or “asset* based” or “strength*-based” or “strength* based” or “asset* mapping” or “strength* mapping” or “appreciative inquiry” or “appreciative inquiries”)

**Table 2 ijerph-16-03778-t002:** Included texts.

First Author, Year, Title	Place	Project Aim	Results	Method
Ager, R.D. et al. (2008) [33] The Youth Video Project: An Innovative Program for Substance Abuse Prevention	New Orleans, USA	A pilot intervention to connect children with positive role models, enhance community ownership of drug issues, empower the community, use community capacities to lay the groundwork for a longer-term program sustained by the community. By developing a video, youth were to learn about healthy attitudes and behaviors related to substance abuse.	Six activities were identified as critical to the program’s success: Family involvement, community engagement, adapting drug education content to fit community characteristics, using a video camera as a vehicle for community field assignments, and evaluation-based learning.	Qualitative interviews and observation of processes relating to principles of capacity building and cognitive dissonance theory. The quantitative questionnaire, paired *t*-test with seven participants to test changes in drug attitudes, perceptions, behaviors and knowledge.
Andresen, W.R. (2011) [40] Using an Asset-based Community Development Initiative to Attract and Retain Young People	Northern Wisconsin and Michigan, USA	To use ABCD in a small community to attract and retain young people and reverse the region’s population decline.	During the asset mapping phase, few young people indicated that they planned to stay in the region. After their involvement in the project, 78% of participants (all ages) stated that their perception of the community had improved. The project led to a vision for, and steps towards, realizing a nature trail infrastructure, a promotional website and a social media campaign. Continued evaluations will assess population decline changes.	Quantitative survey to identify community features that respondents (668 teenagers and professionals under 40 years) considered to be assets. The author observes and describes the community-based initiative that followed to promote, strengthen and connect young people to the identified assets.
Andresen, W.R. (2012) [41] Evaluating an asset-based effort to attract and retain young people.	Northern Wisconsin and Michigan, USA	Evaluation of the use of ABCD in a small community to attract and retain young people and reverse the region’s population decline.	A heightened understanding of local assets led to momentum for their further development, including investments in nature trails and information promotion. These short- and mid-term program results may positively influence long-term population numbers.	Evaluation of the effectiveness of a community-based initiative by qualitative and quantitative methods. Measurement of short-term changes in learning and mid-term changes in action.
Baker, I.R. et al. (2007) [38] An asset-based community initiative to reduce television viewing in New York state.	Rural upstate New York, USA	Catalyzing an established asset-based community partnership to support efforts to reduce television viewing. Part of a broader 3-year study to reduce childhood obesity among rural preschool-aged children.	Asset mapping and focus groups led to identifying desirable actions before planning and implementation of activities in two TV turn-off weeks in 2004 and 2005. Forty indoor and outdoor activities for pre-schoolers and families were provided in public venues. A community-sourced action plan can lend strength to childhood obesity interventions and other public health initiatives.	Case study. Observation and documentation of asset mapping of individual and community strengths followed by focus groups to identify desirable actions. The project engaged childcare staff, administrators and directors from 10 childcare facilities serving on average 276 pre-schoolers.
Hufford, L. (2009) [42] Applying asset-based community development in resident education	Sacramento, California, USA	Communities and Physicians Together (CPT) at the University of California, has developed a service-learning program teaching ABCD since 1999. Their aim has been to establish collaboration with local communities and enable residents (pediatricians) to become effective community advocates. CPT aims to emphasize identifying and utilizing strengths and building capacity, in contrast to the traditional medical model, which emphasizes deficits and needs.	CPT became a well-established partnership between a pediatric residency program, five community collaboratives located in diverse neighborhoods, and a grassroots child advocacy organization. The CPT curriculum teaches residents to build partnerships with their assigned community throughout three years of training, following the ABCD principles. Residents perform activities designed to provide them with a community member’s perspective and partner with communities to implement a project to improve the health of children in that community. The article provides examples of successful community projects.	Case reports and description of the development of the service-learning program teaching ABCD. Refers to reports from students (resident pediatricians) who participated in the program and a qualitative evaluation that demonstrated residents’ attitudes of their role as pediatricians in the community changed with CPT.
Blitz, L.V. et al. (2018) [34] Akuluakulu? Sapasidwa Kanthu (Grown-ups? They Get Nothing): Informing an International Community-University Partnership in Malawi	Rural Malawi	To apply ABCD to establish a university-assisted component of the Malawi Children’s Mission.	ABCD was used as an engagement approach and a set of strategies to identify and mobilize community assets to support guardian that supported orphaned children. Whereas, the project is still in an early stage, an independent network of adults who are invested in the healthy growth of their communities is taking form, and children are, so far, a target group.	Case study—150 orphan children at the Malawi Children’s Mission and their guardians from three rural villages. Interviews and group meetings with community members, village chiefs and staff at the Malawi Children’s Mission.
Johnson Butterfield, A.K. et al. (2016) [14] “Now I know my ABCDs”: Asset-Based Community Development with School Children in Ethiopia.	Addis Ababa, Ethiopia	An adult facilitator aimed to assist children who explored how they could advance the quality of life in their communities. The children had participated in an adult-led ABCD project requested their own ABCD process for their low-income community. The project aimed to explore how ABCD methods can address gaps in community assets for children, to produce knowledge for social work educators, students, and practitioners working to involve children in community partnerships.	The children elected their own facilitators, came to appreciate their own and their community’s strengths and formed communities of learning. Using drawings, flip charts, narratives, sociodrama, poems, art and songs, they presented their findings, including local government and school officials. They pinpointed strengths, assets, gaps and solutions, and gained support for initiatives, including a children’s theatre group that enacts sensitive and challenging themes, and outreach to poverty-struck elderly residents.	Case study. The project process was observed and documented, throughout the three-year period, where the number of participants doubled to 100 children aged 7–14. The study documented the process of various subprojects, i.e., involving language development, performing arts and community service.
Gulley, T. (2006) [35] Building community capacity in southwest Virginia.	Southwest Virginia, USA	The project aimed to increase social capital in a neighborhood after “Ms. G”, who led the project, introduced the idea and wanted to beautify a wall. The project aims to illustrate how community assets can be recognized, and social capital can be increased.	Children aged 4 to 17 participated in painting elements from community history on a wall, with support from the fire department (wall cleaning), parents (snacks, transport), the town council (funds), a student (mural painting experience). The author concludes that the rural community residents achieved their goal, with the painted mural as a reminder of a positive experience. Implications for nursing education and community partnerships are discussed.	Description of a case intervention.
Bowns, C. (2011) [43] Facilitating the Production of Place-Based Knowledge for Participatory Community Development in Rural Pennsylvania	Rural Pennsylvania, USA	Research questions: “How can local communities contribute to place-sensitive development in rural places?” and “What are the outcomes of adult and child participation in community-based projects?”.	Local communities were engaged in identifying and prioritizing local opportunities for revitalization and invited to comment on proposals. Place-based knowledge can benefit community improvement through participatory processes.	Case study. Description of service-learning projects in three rural communities, delivered by landscape architecture students.
Pan, R. et al. (2005) [37] Building Healthier Communities for Children and Families: Applying ABCD to Community Pediatrics	Sacramento, California, USA	The service-learning program aimed to apply ABCD to community pediatrics. In this concrete case, the aim was to involve community partners through ABCD to address the problem of a high frequency of children arriving at the emergency ward bitten by dogs.	The chosen case illustrates that ABCD can be used successfully in the service-learning program. In this case, the asset mapping exercise involved dog owners, crossing guards, children, a neighborhoods association, an elementary school and a city park. The resident and community partners organized a dog-safety fair, where dog owners allowed children to practice dog-safety behavior. New social networks and social norms were created in the community. The authors observe that this approach is likely to reduce dog bites.	Case study on the application of ABCD based on the report from a pediatric resident.
Roberts, S. et al. (1998) [39]	Edmonton, Canada	The Community Development Office of Capital Health (CDO) Westlawn Junior High School aimed to involve students and community groups in development processes by the use of ABCD. They aimed to discover, connect, and mobilize students’ assets and gifts and connect these with those of people and groups in the communities, and to evaluate the project.	Fifteen grade 9 students met for two full-day workshops and 14 afternoons, in addition to some full days spent on special events and projects. Assisted by facilitators the youth made two projects: Flower planting with the community and carnival for daycare children. The children developed their participatory skills. The evaluation recommended: (1) Securing funding for continuation. (2) Pre and post-test to measure changes in students’ self-esteem and leadership abilities. (3) Include business plans as part of the projects. (4) Increase the involvement of external partners. (5) and of the parents and staff. (6) Modify daily plans. (7) Daily reflections and evaluations. (8) Keep records of expenses and human resources.	Project description and evaluation-based on objectives set before the project started, followed by observations documented by the participants. Unpublished report.
Shah, R.W. et al. (2018) [36] Fostering eABCD: Asset-based community development in digital service-learning	Lincoln, Nebraska, USA	To explore how aspects of asset-based com-munity development can be enacted in online asset-based community development (eABCD).	The study of a digital writing partnership between college students and rural youth in the ninth grade. It illustrates how students can be supported in asset-based, relationship-driven, and internally focused interactions in online service-learning collaborations.	Case study based on online text dialogues. Texts from one term were coded using Dedoose to identify aspects of the three ABCD principles. Community partners and youth participants also answered questions geared towards the ABCD principles.

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
