# Peer review of "Can Asset-Based Community Development with Children and Youth Enhance the Level of Participation in Health Promotion Projects? A Qualitative Meta-Synthesis"

_ijerph, 2019, doi:10.3390/ijerph16193778_

Round 1

Reviewer 1 Report

Congratulations for this very intersting and important work.

I suggest to improve the quality of the work:

a) Make a more clean and simple title. Instead of a question maybe you could identify the analyzed variables:

suggestion: "The relation bettween the use of ABCD model with children, youth and shools and the level of participation in health promotion projects- A
qualitative meta-synthesis"

b) there are too many key words. Six would be enough. Try use terms included in Mesh.

c) In the refereces along the text, when you have a citation change for e.g. : (7 p.11­-14) to (7:11-14)

d) maybe you could work on figure 2 (pág 12) to make it more clear. Instead of coping it you could built it in the text, its an easy scheme to do.

Author Response

Thank you for your thorough reading and for your kind comments and constructive suggestions to improve the work.

a) We have realised that the title is long, and taken out the schools, as schools are not central agents in the projects.

b) We would like to reduce the number of key words to six. I will have to write to the editor to find out how we technically can do this. To use Mesh terms is a good suggestion.

c) The references to pages were not correct, as you pointed out. We have now followed the guidelines for MDPI in line with the standard for the journal. 

d) Thank you for the comment on the figure 2. With your comment we realized that it comes to abruptly and should be introduced with a short text. Now we have added and introduction and explain that it is an illustration to the text under the paragraph "Degrees of adherence to ABCD principles and levels of participation" in addition to providing some more introduction under the paragraph on "Classification of participation".

Reviewer 2 Report

1. The manuscript proposes an interesting revision regarding the ABCD in the framework of the relations between youth, their culture and the environment in which they operate.

2. Without being a bibliometric work, it solves the conceptual tendencies in studies that are oriented to mass culture and media, which is not present in the subsequent discussion.

3. Understanding that the evidence is in methodological trends, one could also add a look at the more relational issues in ABCD strategies, such as community representation and its development projections. This is mentioned on pages 13, 14 and 15.

4. Finally, if early projects impact on health policies, a closing reflection that indicates which one or more of these strategies is more valid and if the difference in location (global or global north) impacts the design of strategies like this would be interesting.

5. The work has interesting elements and can be improved with a new reading about its own results.

Author Response

Thank you for your time and for your kind comments and constructive suggestions to improve the work.

We have revised the manuscript and tried to follow up on your suggestions.

We find your comment on possible impact on health policies interesting. We have integrated this, but unfortunately there is so far not research to build on related to impact on policies. The systematic research on Asset Based Community Work in health promotion strategies are still in an early phase, although rapidly gaining momentum through rapid increase of new works in the latest two years.

The discussion have been revised, as some of it has been moved to results, and other parts developed.

Reviewer 3 Report

This manuscript aims to review evidence regarding the utility of the ABCD approach in promoting children’s, youths’, and students’ participation in health promotion initiatives. At its heart, this paper addresses important tenets of equity and inclusion in public health; young people have agency in their communities and deserve ample opportunities for civic participation. However, the manuscript in its current form does not seem to clearly focus on environmental science or public health priorities in a way that makes sense for publication in IJERPH.

Introduction:

The ‘Introduction’ states an important point that community participatory approaches are integral to eventually mitigating health inequities. However, the section should be better utilized in framing the paper as a whole. The ABCD approach needs to be briefly explained/summarized in the beginning of the paper. Including a brief explanation of the approach in the Introduction would help the reader orient to the manuscript and its goals. What is the ABCD approach? Why is it particularly relevant to addressing public health inequities? Further, why is it an especially relevant approach for engaging young people? What previous research has been done on this topic, and why is this review paper warranted? What is the gap in knowledge that this manuscript is filling? The first small paragraph is confusingly placed; it seems to be better suited toward the end of the Introduction section after the authors have presented the broader scope of their manuscript topic. Also, the aims stated in those first sentences do not match the abstract’s stated focus on public health promotion strategies. The ‘Theoretical approach’ section appears more so to present an evaluation framework for the authors’ findings in this paper rather than to discuss theoretical concepts that undergird why children should be considered valuable actors in ABCD approaches to health promotion projects, and why such approaches would enhance youths’ participation. It seems reasonable to utilize Hart’s model of children’s participation as an evaluative tool, but this information may be better suited to the preceding section describing general literature on ABCD. Relatedly, that preceding section (‘Two decades of Asset-Based Community Development’) seems to focus almost exclusively on the history of ABCD and what it is. This is important information, but it would behoove the authors to also review some of the literature that their current manuscript hopes to synthesize: namely, previous studies or papers that describe applications of ABCD to youths’ participation in public health promotion projects. Presenting some of this evidence might help the authors re-iterate why a synthesis of such literature is needed and thus why their manuscript addresses an important gap in the public health literature. This first part of the paper also needs to clarify why young adults are included as a population in the review despite most of the manuscripts’ focus landing on children. For example, are there specific concepts behind the tenets of ABCD that apply coherently to children and college students alike?

Methods:

For clarity, the authors should state what their explicit inclusion and exclusion criteria were prior to referring to their general use of inclusion or exclusion criteria. It would be helpful if the authors referred the reader to Figure 1 as well as mentioned their use of the PRISMA guidelines. For transparency and replicability, it should also be stated what, exactly, the “blinded process” was for citation review. The 'Methods' lack clarity and detail and don’t seem congruent with the overall aim of the paper. How were data extracted by the two reviewers? Was a standardized form used by both reviewers so that they were extracting the same data for their decision-making regarding inclusion or exclusion? It is notable that the terms “appreciative inquiry” and “appreciative inquiries” were included as key search terms. While certainly related, appreciative inquiry and ABCD are considered separate approaches to engaging with communities. Why was this a key part of the authors’ search for literature? These terms don’t appear elsewhere in the manuscript, and their inclusion needs clarified. The authors’ stated aim entails examining how the ABCD approach has been used to engage young people’s participation in public health promotion strategies. However, the inclusion/exclusion criteria do not address public health problems. (For example, what was the public health concern in Gulley’s [2006] netted study that the authors included? How is young people’s perception of their community, and their proclivity to stay in the community, a health promotion project?)

Results:

Where does “a total of 111 texts on ABCD” come from? The authors’ PRISMA diagram presents 119 studies assessed for eligibility after the removal of 2,925 citations not about ABCD projects. The bulk of the ‘Results’ section is structured like an annotated bibliography. Because the findings of each included study were listed by citation rather than synthesized in an integrative way, it’s not clear what key constructs emerged in this specific area of the literature. Related to the above point: if a primary goal of this study is to evaluate young people's participation in ABCD-framed public health promotion projects and each project’s adherence to ABCD principles, then much of the ‘Results’ section should be dedicated to these findings. The summaries of each study, as currently presented in an annotated bibliography-style fashion, can be substantially condensed. The ‘Results’ seem to be framed by stage of child development but then the final section reflects a structural type (“service-learning”) of project. Since the projects reviewed in that section seem to pertain to young adults in higher education, why not keep a developmental subsection title? The ‘Results’ section, like others, includes some extended block quotations from other authors. Direct quotations should be used sparingly, and only when the original source contains articulated ideas or concepts that truly cannot be better or adequately expressed in any other words but the original author’s own. Rather than presenting block quotations from the sources the authors draw upon, the authors should interpret and integrate the concepts and ideas behind those quotations into their own words and synthesis.

Discussion:

Much of the ‘Discussion’ section is dedicated to presenting findings from the meta-synthesis and thus should be presented in the 'Results' section. The authors’ findings regarding how adherent each included citation was to ABCD principles comprise results and thus should be in the ‘Results’ section. Elaboration on the EPICURE elements belongs in the ‘Methods’ section, not in the ‘Discussion’ section. The ‘Discussion’ does not heavily draw upon the authors’ interpretations of the broad arcs of findings across studies. What are the practical and research implications for so few included citations adhering to certain ABCD principles? How does this inform the authors’ conception of how future health promotion projects should interface with communities and children? Rather than summarizing McKnight and colleagues’ work again in such detail, the authors should allude to McKnight et al.’s guiding work in discussing conceptual implication of the current study.

Author Response

Dear reviewer,

Thank you for the thorough reading and the valuable comments. We have revised the manuscript based on your suggestions. New parts is marked with yellow, and headlines are marked with yellow for paragraphs were we merely have made the text shorter and hopefully clearer.

We realized that the aim of the paper has not been clear, and the introduction is therefore new. With the new introduction the relevance for the journal is clearified.

The paragraph "Theoretical approach" has been changed. The information about the development of ABCD has been extended, linked to development related to health promotion and we present some of the discussion and critique of ABCD. Under the new heading "Classification of participation", we introduce the participation ladder and other perspectives on the participation of children and youth, and why it is worth attention in health promotion initiatives.

We have now made it clear that the texts that involved university students are included because the students worked with children in health promotion projects. We apologize for not presenting this in the first place.

We also did a mistake in the Methods section when presenting two different numbers for the texts on ABCD. This is now corrected. The mistake happened when we updated the search and added the new findings, but missed to change a number in the figure. We have also added more information in this section, and explained the relevance of "appreciative inquiry". We did not perform a citation review, but merely checked the reference lists in the included text to search for additional texts on youth or children in ABCD projects.

The results section have been restructured, according to your suggestions. Some of the elements that we had placed under discussion have also been moved to the results section. Block quotations have been used more sparingly.

The discussion is now linked to implications for health promotion and less to the the work of McKnight et al. The discussion of the scientific quality (EPICURE) has been moved to the methods section.

We hope that you will find that your comments have found fertile ground, and that you will find that the manuscript is improved.
